# Persulfide-Responsive Transcription Factor SqrR Regulates Gene Transfer and Biofilm Formation via the Metabolic Modulation of Cyclic di-GMP in *Rhodobacter capsulatus*

**DOI:** 10.3390/microorganisms10050908

**Published:** 2022-04-26

**Authors:** Takayuki Shimizu, Toma Aritoshi, J. Thomas Beatty, Tatsuru Masuda

**Affiliations:** 1Graduate School of Arts and Sciences, The University of Tokyo, Tokyo 153-8902, Japan; aritoshi-tohma403@g.ecc.u-tokyo.ac.jp (T.A.); cmasuda2@g.ecc.u-tokyo.ac.jp (T.M.); 2Department of Microbiology and Immunology, University of British Columbia, Vancouver, BC V6T 1Z3, Canada; j.beatty@ubc.ca

**Keywords:** transcriptional regulation, gene transfer, persulfide, redox signaling, cyclic GMP

## Abstract

Bacterial phage-like particles (gene transfer agents—GTAs) are widely employed as a crucial genetic vector in horizontal gene transfer. GTA-mediated gene transfer is induced in response to various stresses; however, regulatory mechanisms are poorly understood. We found that the persulfide-responsive transcription factor SqrR may repress the expression of several GTA-related genes in the photosynthetic bacterium *Rhodobacter capsulatus*. Here, we show that the *sqrR* deletion mutant (Δ*sqrR*) produces higher amounts of intra- and extracellular GTA and gene transfer activity than the wild type (WT). The transcript levels of GTA-related genes are also increased in Δ*sqrR*. In spite of the presumption that GTA-related genes are regulated in response to sulfide by SqrR, treatment with sulfide did not alter the transcript levels of these genes in the WT strain. Surprisingly, hydrogen peroxide increased the transcript levels of GTA-related genes in the WT, and this alteration was abolished in the Δ*sqrR* strain. Moreover, the absence of SqrR changed the intracellular cyclic dimeric GMP (c-di-GMP) levels, and the amount of c-di-GMP was correlated with GTA activity and biofilm formation. These results suggest that SqrR is related to the repression of GTA production and the activation of biofilm formation via control of the intracellular c-di-GMP levels.

## 1. Introduction

Bacteria do not perform sexual reproduction but are capable of acquiring exogenous DNA by horizontal gene transfer (HGT), which is important for genetic diversity and evolution [1]. Although HGT is classically mediated by transformation, conjugation and transduction, small bacteriophage-like particles called gene transfer agents (GTAs), which package random segments of cellular DNA, also mediate HGT [2,3]. GTA was originally discovered as a novel genetic vector in the alphaproteobacterium *R. capsulatus* [4], and subsequently in other diverse bacteria [5]. In particular, homologous GTAs are conserved in a number of families in the alphaproteobacterial order *Rhodobacterales*, which occupy over 25% the total prokaryotic community in some marine environments [6,7,8]. Moreover, gene transfer rates of antibiotic resistance markers via putative GTAs were, remarkably, one-million-fold higher than previous estimates of transformation and transduction rates in natural environments [9]. Therefore, it appears that GTAs are widely employed as an important genetic vector in bacterial HGT.

The GTAs of *R. capsulatus* (RcGTA) have been investigated in detail. It has been reported that gene transfer by RcGTA is induced at the stationary phase and by carbon starvation [10,11]. In these regulations, the GtaI-GtaR quorum-sensing system [12,13] and the CckA-ChpT-CtrA phosphorelay [14,15] are centrally important. Moreover, the transcription factor GafA functions as a direct activator of the RcGTA structural gene cluster [16]. Quorum-sensing is a well-known cell-to-cell communication system in bacteria. Bacteria produce and release autoinducers such as acyl-homoserine lactones (AHL) during growth, sense the outside concentration of autoinducers to recognize high concentrations of related species, and consequently modulate the expression of certain genes [17]. *R. capsulatus* possesses an AHL synthase (GtaI), and its regulator (GtaR) indirectly regulates RcGTA production in response to AHL [13]. This regulation mechanism is thought to be involved in RcGTA production associated with the growth phase. The CckA-ChpT-CtrA phosphorelay is one widely studied phosphorelay that controls the cell cycle in many alphaproteobacteria. CtrA functions as a response regulator whose activity is controlled through the histidine kinase CckA [18] and the histidine phosphotransferase ChpT [19]. This phosphorelay system also regulates RcGTA production, switching from the production phase to the release phase dependent on the phosphorylation state of CtrA [14,15]. The PAS domain protein DivL promotes the phosphorylation of CtrA via the enhancement of CckA kinase activity, and thereby RcGTA production is controlled [20]. The RNA polymerase omega subunit is required for the RelA/SpoT-related stringent response induction of RcGTA production under carbon starvation, although its association with the CtrA pathway is unclear [11]. Another aspect of the CtrA regulation of RcGTA production is manifested by c-di-GMP, via the possible modulation of c-di-GMP metabolic enzymes [21]. Thus, the outline of a complex regulatory network has been revealed; however, the details of each regulatory pathway and their possible connections are unclear.

Recently, it was reported that persulfides, which are oxidized sulfur species generated from sulfide, are signaling molecules and modulate various physiological functions in both prokaryotes and eukaryotes [22,23,24,25,26,27]. Since hydrogen sulfide was abundant over oxygen in the prebiotic Earth period, and for much of the Archean Eon, it is considered that sulfide and persulfide contributed significantly to evolution in terms of energy metabolism and signal transduction [28]. We have identified the persulfide-responsive transcription factor SqrR as a regulator of sulfide-dependent photosynthesis in *R. capsulatus* [22]. SqrR binds to the promoter regions of target genes to repress their expression in the absence of persulfide. In the presence of persulfide, SqrR forms an intramolecular tetrasulfide crosslink between two cysteine residues and loses the ability to bind DNA, which results in the de-repression of target genes [22]. SqrR is the master regulator of the persulfide response: RNA-seq analysis has shown that contributions of SqrR regulate 45% of sulfide-responsive genes [22]. Interestingly, RNA-seq data have indicated that several RcGTA-related genes, such as the RcGTA capsid protein gene (rcc01687), *chpT* (rcc03000) and *divL* (rcc00042), were up-regulated between three and nine-fold in an *sqrR* deletion mutant (Δ*sqrR*) [22] (Table 1). Based on this observation, it appears that SqrR could contribute to the regulation of RcGTA production. Here, we provide evidence that SqrR regulates RcGTA production via H_2_O_2_ signaling. Because SqrR controls intracellular c-di-GMP levels, we suggest that SqrR modulates both RcGTA production and biofilm formation.

## 2. Materials and Methods

### 2.1. Bacterial Strains, Media, and Growth Conditions

*R. capsulatus* strains B10 [29], DE442 [30] and mutants were grown under aerobic–dark (aerobically) or anaerobic–light (photosynthetically) conditions at 30 °C in PYS or YPS, a rich medium, or RCV, a minimum medium [29,31,32]. For photosynthetic growth, a light-emitting diode (λ_max_ = 850 nm) (CCS) was provided. To establish anaerobic conditions, cultures in screw-capped test tubes were almost completely filled with the medium. Gentamycin and rifampicin were used at a concentration of 1.5 µg/mL and 75 µg/mL, respectively.

*Escherichia coli* strains were grown on Luria Bertani (LB) medium at 37 °C. Ampicillin and gentamycin were used at a concentration of 100 µg/mL and 10 µg/mL, respectively.

### 2.2. Cloning and Mutagenesis

*sqrR* disruption in *R. capsulatus* DE442 was performed using the plasmid pZJD29a::Δ*sqrR*, as previously described [22]. For the disruption of the *oxyR* gene, two ~500 bp DNA fragments consisting of the N-terminal and C-terminal regions of *oxyR* were amplified by polymerase chain reaction (PCR) with KOD one polymerase (TOYOBO). Two sets of primer pairs were used for the amplification; one was a forward primer, oxyR F1, and a reverse primer, oxyR R1, and the other was a forward primer, oxyR F2, and a reverse primer, oxyR R2 (Table 2). These two fragments were cloned into the *Bam*HI-cut pZJD29a [33] by an In-Fusion HD Cloning kit (Clontech, Mountain View, CA, USA). The obtained plasmids were introduced into *R. capsulatus* strains by conjugation with the *E. coli* strain S17-1/*λpir*, and the subsequent homologous recombination events were induced as described [33]. The isolated mutants were analyzed by DNA sequencing to confirm a deletion.

### 2.3. GTA Transduction Assay

GTA transduction assays were performed referring to previous study [34]. The rifampin-resistant donor strain was grown photosynthetically to the log phase (WT: OD_660_ = 1.2, DE*sqrR*: OD_660_ = 1.1) or stationary phase (WT: OD_660_ = 1.7, DE*sqrR*: OD_660_ = 1.4) in YPSm medium. Then, 1.5 mL culture normalized at OD_660_ = 1.0 with 20 mM Tris-HCl (pH 8.0) was centrifuged, and supernatant was filtered using a 0.45 μm membrane filter. Next, 100 μL of the obtained sample was mixed with 500 μL of rifampin-sensitive B10 recipient cells grown photosynthetically to mid log phase (OD_660_ = 0.5). The harvested cells were re-suspended with G buffer (10 mM Tris-HCl (pH 8.0), 1 mM MgCl_2_, 1 mM CaCl_2_, 1 mM NaCl, 250 μg/mL BSA). The mixture was incubated under aerobic shaking conditions at 30 °C for 1.5 h to undergo gene transfer to the recipient cells. The harvested cells were re-suspended with RCV medium and plated on rifampin-containing plates. The number of rifampin-resistant colony-forming units was determined.

### 2.4. Western Blotting of Capsid Protein

*R. capsulatus* was grown photosynthetically to the log phase or stationary phase in YPSm medium. Then, 1.5 mL culture normalized at OD_660_ = 1.0 with 20 mM Tris-HCl (pH 8.0) was separated by centrifugation, and the supernatant was filtered through a 0.45 μm pore-size membrane filter. Cell pellets were re-suspended in 500 μL of 20 mM Tris-HCl (pH 8.0) and disrupted by sonication. Proteins were then separated by 15% SDS-PAGE gels. After electrophoresis, proteins were blotted onto PVDF membrane and probed with commercially available RcGTA major capsid antiserum (AS08 365; Agrisera AB) according to the product’s instructions. The secondary antibody was visualized by Clarity Western ECL substrate (Bio-Rad, Hercules, CA, USA).

### 2.5. RNA Isolation and Quantitative Real-Time PCR (qRT-PCR)

*R. capsulatus* was grown photosynthetically to the log phase or stationary phase in YPSm medium. For sulfide or H_2_O_2_ treatment, a final 0.2 mM of Na_2_S or 1 mM H_2_O_2_ was added at the mid-log phase (OD_660_ = 0.7), and cells were further grown for 30 min. Then, 0.5 mL of cells were harvested, and the total RNA of each sample was extracted using NucleoSpin RNA (TaKaRa). The quality of the purified RNA was checked to confirm a typical OD_260_ to OD_280_ ratio of approximately 2.0. RNA samples were reverse transcribed using a PrimeScript RT Reagent kit (TaKaRa, Shiga, Japan). qRT-PCR reactions and detection were performed using the THUNDERBIRD Next SYBR qPCR Mix (TOYOBO, Osaka, Japan) and the CFX Connec Real-Time System (Bio-Rad). As an internal control, the house-keeping gene *uvrD* that encodes DNA helicase [16] was used with the gene-specific primers (Table 2).

### 2.6. Overexpression and Purification of SqrR

Recombinant SqrR was overexpressed in the *E. coli* strain BL21 (DE3) overexpression system utilizing pSUMO::SqrR plasmid, and was purified as previously described in [22].

### 2.7. Gel Mobility Shift Analysis

An FITC-labeled 200–300 bp DNA probe containing each promoter region was prepared by PCR amplification with the primer sets (Table 2). The amplified fragment was cloned into a pUC19-linearized vector (TaKaRa) using the In-Fusion HD Cloning kit (Clontech). The inserted DNA was amplified by PCR with a FITC-labeled primer, as described previously [35]. The amplified DNA was purified and utilized as a probe for the assays.

The gel shift assay was performed as described in [22]. The binding reaction buffer (25 mM Tris-HCl (pH 8.0), 100 mM NaCl, 2 mM MgCl_2_, 6% glycerol, 0.5 mM DTT and 50 µg/mL heparin) was utilized for the reaction with the DNA probes and proteins. For the electrophoresis, 7% polyacrylamide gel was used in a buffer composed of 25 mM Tris-HCl (pH 8.0), 1 mM ethylenediaminetetraacetic acid (EDTA) and 144 mM glycine. After electrophoresis, the gel was analyzed using the Lumino Graph I (ATTO, Tokyo, Japan).

### 2.8. Quantification of c-di-GMP

*R. capsulatus* was grown photosynthetically to the log phase or stationary phase in YPSm medium. Cells from 1 mL culture were disrupted by sonication with 20 mM Tris-HCl (pH 8.0). After centrifugation, the supernatant was utilized for detecting c-di-GMP using the Cyclic di-GMP ELISA Kit (Cayman). The obtained values were normalized by the protein concentration of the supernatant.

### 2.9. Quantification of Biofilm Formation

The amount of biofilm formation was quantified based on crystal violet staining by the Biofilm Formation Assay Kit (DOJINDO, Kumamoto, Japan), according to the product’s instructions. In short, *R. capsulatus* was grown photosynthetically to the stationary phase and dispensed onto a 96-well plate. The plate was covered by a 96-well peg lid containing protrusions that were immersed into the culture of the wells. Biofilm formed around this protrusion during overnight cultivation at 30 °C. The 96-well peg lid was washed with 20 mM Tris-HCl (pH 8.0) and the formed biofilm was stained by crystal violet solution. This staining was eluted by 100% ethanol after being washing twice with 20 mM Tris-HCl (pH 8.0). The absorbance at 600 nm of eluate was measured using the GloMax Multi Detection system (Promega, Madison, WI, USA). The obtained values were normalized by the OD_660_ of the culture.

## 3. Results

### 3.1. SqrR Contributes to GTA Production and Release

Because only ~3% of the cells in a population are responsible for the release of RcGTA in the WT *R. capsulatus* strain SB1003 [36,37], we constructed the Δ*sqrR* mutant using the *R. capsulatus* RcGTA overproducer strain DE442 as a background to verify whether SqrR contributes to RcGTA production and release. The DE442 strain had a mutation in the rcc00280 gene that increased the amount of RcGTA production [37], but was WT in terms of the *sqrR* gene. The growth rates of the parental wild-type strain (WT) and Δ*sqrR* were similar, although Δ*sqrR* showed a slightly longer log growth phase (Figure 1A). Both strains were cultivated using the YPSm-rich medium, and gene transduction was analyzed at the log and stationary phases to explore whether the deletion of *sqrR* affected the functional activity of RcGTA. In the WT strain, there was a low frequency of gene transduction in the log phase, with an induction in the stationary phase (Figure 1B). In contrast, the Δ*sqrR* mutant showed a significantly greater gene transduction frequency in both the log and stationary phases than the WT strain. We also measured the amount of RcGTA capsid protein by Western blotting using an anti-capsid antibody. A greater amount of RcGTA capsid protein was detected in Δ*sqrR* compared to the WT strains of both the pellet (cellular) and supernatant (extracellular) fractions at the stationary phase (Figure 1C). Although it was difficult to detect a difference at the log phase because of the weakness of the signal, the gene transduction frequency clearly showed a higher amount of RcGTA released from the Δ*sqrR* than the WT strain at the log phase (Figure 1B). These data indicate that SqrR contributes to RcGTA production and release.

We subsequently analyzed the transcript levels of RcGTA-related genes by real-time PCR (qRT-PCR). The transcript of the gene encoding the capsid protein was increased by *sqrR* deletion in correlation with the intracellular amount of GTA capsid protein at the log phase (Figure 2A). Moreover, the *gafA* gene, which encodes a direct activator of the RcGTA gene cluster, was also up-regulated in the Δ*sqrR* mutant. These data indicate that SqrR is needed to repress these genes at the log phase. In contrast, the transcripts of the phosphorelay-related regulators, *divL*, *cckA*, *chpT,* and *ctrA*, were unaffected (Figure 2A). We further tested the effect of sulfide on these transcriptional changes, because SqrR functions as a (per)sulfide-responsive transcriptional repressor. Unexpectedly, treatment with sulfide did not alter these transcript levels at the log phase of either the WT or Δ*sqrR* strains (Figure 2B). Since SqrR senses sulfide by persulfidation via the oxidation of cysteine residues, it was possible that another oxidant, such as hydrogen peroxide (H_2_O_2_), could react with SqrR in vivo. Therefore, we subsequently tested the effect of H_2_O_2_ on the transcript levels of RcGTA-related genes. It was found that transcripts of the capsid and *gafA* genes were increased by the treatment of cells with H_2_O_2_, and this up-regulation did not occur in the Δ*sqrR* mutant (Figure 2C), indicating that SqrR contributes to the H_2_O_2_-induced transcription of RcGTA-related genes at the log phase. We also analyzed the transcript levels when cells were treated with mitomycin c, which induces the SOS response via DNA damage [38], in order to explore whether this effect of H_2_O_2_ on the transcript levels was due to a change in the intracellular redox state, or if it was an SOS response induced by H_2_O_2_ [39] that could be related to RcGTA regulation [40]. Although the *gafA* transcript was increased approximately two-fold by treatment with mitomycin c in the WT strain (Figure 2D), this change was a lot smaller than that resulting from the treatment with H_2_O_2_, indicating that the SqrR-related H_2_O_2_-induced transcriptional change in the RcGTA-related genes may have been due to a change in intracellular redox state.

### 3.2. Molecular Mechanism of the SqrR-Related H_2_O_2_-Induced Transcription of GTA-Related Genes

To verify whether SqrR directly regulates RcGTA-related genes, we performed gel mobility shift assays using SqrR recombinant protein and DNA probes containing the predicted promoter region of each gene. Unexpectedly, SqrR did not bind to the promoter regions of the *gafA* or capsid gene (upstream of the first gene of the structural gene cluster, g1). However, distinct binding to the *sqr* promoter region as a positive control was shown (Figure 3A). Therefore, the SqrR modulation of the RcGTA production appeared to require at least one additional factor. It is well known that the transcription factor OxyR functions as a master regulator of reactive oxygen species (ROS) signaling in bacteria [41,42]. As *R. capsulatus* also has OxyR, we explored the contribution of OxyR to the H_2_O_2_-induced transcription of RcGTA-related genes. To elucidate this, we constructed the *oxyR* single-deletion mutant (Δ*oxyR*), the *sqrR,* and the *oxyR* double-deletion mutant (Δ*sqrR*Δ*oxyR*), and measured the transcript levels of capsid and *gafA* after treatment with H_2_O_2_ (Figure 3B). The transcripts were slightly increased in the Δ*oxyR* strain compared to the Δ*sqrR* strain and, particularly, the transcript of the capsid protein significantly increased. Moreover, the Δ*sqrR*Δ*oxyR* double mutant showed significantly lower levels of these transcripts than every single mutant, except for the transcript of the capsid in Δ*sqrR*. These results indicate that SqrR and OxyR independently regulate the H_2_O_2_-induced transcription of RcGTA-related genes. Overall, SqrR appears to act downstream of the H_2_O_2_-induced regulation of RcGTA production, independently of OxyR, and indirectly mediates the transcriptional regulation of RcGTA-related genes to control RcGTA and gene transfer frequency.

It has been reported that CtrA regulates the transcript levels of c-di-GMP metabolic enzymes (rcc00620, rcc00645, rcc02629 and rcc02857), and RcGTA production is negatively regulated by c-di-GMP [21]. Interestingly, our previous RNA-seq data indicated a contribution of SqrR to the expression of rcc00645 and rcc02857, which contain both GGDEF (diguanylate cyclase) and EAL (phosphodiesterase) domains [22] (Table 1). To explore the effect of SqrR on the level of c-di-GMP, we compared the intracellular amount of c-di-GMP between the WT and Δ*sqrR* strains. A significant decrease in the amount of c-di-GMP was detected in the Δ*sqrR* mutant compared with that in the WT strain at both the log and stationary phases (Figure 4A). We further investigated whether these changes in intracellular c-di-GMP levels were correlated to the transcript level changes in the c-di-GMP metabolic enzymes. Relative to the WT strain, the transcript of rcc00620, which encodes a c-di-GMP catabolic enzyme, was decreased at the log phase, and that of rcc02629, which encodes c-di-GMP synthase, was increased at the stationary phase in the Δ*sqrR* mutant (Figure 4B). These data do not correlate with a decrease in intracellular c-di-GMP levels. Moreover, SqrR did not bind to the promoter region of rcc00620 and rcc02629 (Figure 4C). These results indicate that SqrR does not modulate intracellular c-di-GMP levels via the transcriptional regulation of CtrA-regulated c-di-GMP metabolic enzymes, but by another of the ~14 possible genes encoding a diguanylate cyclase in the *R. capsulatus* genome.

It is well known that c-di-GMP positively controls biofilm formation [43]. In order to confirm the effect of the intracellular c-di-GMP levels, we analyzed biofilm formation in the WT and Δ*sqrR* strains. Biofilm formation was significantly lower in Δ*sqrR* than in WT (Figure 5). This observation was congruent with the lower intracellular c-di-GMP levels in Δ*sqrR* compared to those in the WT strain (Figure 4A). Thus, the phenotype of biofilm formation activity also supports the regulation of c-di-GMP production by SqrR.

## 4. Discussion

We studied the contribution of SqrR to gene transfer via RcGTA in order to explore the possibility of a novel regulatory process in RcGTA production. We have demonstrated that SqrR transduces the H_2_O_2_-mediated regulation of RcGTA production and biofilm formation by c-di-GMP as a novel regulatory pathway in *R. capsulatus*. This conclusion is based on the effect of *sqrR* deletion on RcGTA production and the transcript levels of RcGTA-related genes. The Δ*sqrR* mutant showed higher gene transfer frequency and greater amounts of intracellular and released RcGTA, as compared with the WT (Figure 1B,C). The transcript levels of the RcGTA capsid protein and the GafA direct activator of the RcGTA gene cluster transcription were up-regulated by *sqrR* deletion, and this regulation mediated H_2_O_2_ signaling (Figure 2). Although previous RNA-seq data have shown that *chpT* and *divL* are also regulated by SqrR (Table 1), these transcript levels were not changed by the deletion of *sqrR* (Figure 2A). This discordance could be due to different growth conditions, aerobic conditions in the previous study, or the anaerobic conditions in this study, because RcGTA production is affected by the oxygen tension of the culture. We note that RcGTA production was delayed from the induction of the transcripts at the log phase. It appears that transcription induction starts in the log phase, but high levels of RcGTA protein do not accumulate until the stationary phase. In addition, SqrR modulated the amount of intracellular c-di-GMP, which induced RcGTA production and inhibited biofilm formation (Figure 1, Figure 4 and Figure 5).

c-di-GMP is a ubiquitous bacterial second-messenger molecule that regulates many bacterial functions and behaviors [44]. It is thought that c-di-GMP affects the CckA-ChpT-CtrA phosphorelay by enhancing the phosphatase activity of CckA, and thereby gene transfer by RcGTA is negatively regulated by nonphosphorylated CtrA [21]. Indeed, gene transfer activity via RcGTA is affected by altering intracellular c-di-GMP levels by the overexpression of the exogenous c-di-GMP metabolic enzyme [21], and CckA phosphatase versus kinase activity is modulated by c-di-GMP [45]. Moreover, in positive regulatory systems of biofilm formation, c-di-GMP allosterically modulates regulators—such as effector kinases in two-component systems or transcription factors—to promote biofilm formation [46,47,48]. Our observations clearly show the alteration of RcGTA production and biofilm formation associated with intracellular c-di-GMP levels (Figure 1C, Figure 4A and Figure 5). Therefore, we suggest that SqrR regulates the levels of enzymes that metabolize c-di-GMP, resulting in changes in the levels of c-di-GMP which, in turn, regulate the levels of RcGTA [21]. It has been reported that c-di-GMP-mediated gene transfer is regulated by a two-component system composed of a sensor histidine kinase encoded by rcc00621 and a c-di-GMP catabolic enzyme encoded by rcc00620 as a response regulator [49]. Our findings indicate a new regulatory pathway of gene transfer via the control of c-di-GMP levels by SqrR. The regulatory network of RcGTA production comprises various response systems and the CtrA-dependent central regulation system to obtain a variety of abiotic stress responses [10,11,12,13,20,21]. Thus, SqrR functions as a redox stress-responsive c-d-GMP modulator, and may be valuable to appropriately regulate RcGTA production via the CtrA phosphorelay in response to one of several abiotic stresses.

SqrR should be basically employed as the persulfide-specific responsive regulator in *R. capsulatus* [22]. The promoter activity of the *sqr* gene, which encodes sulfide:quinone reductase, the transcription of which is repressed by SqrR, was accelerated not by H_2_O_2_, but by sulfide in vivo [22]. Moreover, previous MS-based reactivity profiling has revealed that the modification of cysteine residue results from a reaction with persulfide, but not other oxidants such as H_2_O_2_ [50]. Surprisingly, our data showed a distinct association between SqrR and the H_2_O_2_-induced regulation of RcGTA production and biofilm formation (Figure 2 and Figure 5). It has been reported that the well-established ROS responsive regulator OxyR senses not only H_2_O_2_ via sulfene (–SOH) and/or the disulfide formation of cysteine residue(s), but also persulfide via the persulfidation (–SSH) of cysteine residues [51]. Given that persulfide might have been present earlier than ROS on the prebiotic Earth [28], and that persulfide plays an important role as a signaling molecule for organisms as well as ROS, it is reasonable to suggest that the persulfide sensor protein detects both persulfide and ROS. Therefore, SqrR might have the ability to mediate ROS signal transduction in vivo, although the molecular mechanisms of how SqrR senses or mediates H_2_O_2_ signaling are unclear. One possible mechanism is that the heme-binding ability of SqrR is available for mediating H_2_O_2_ signaling. We have previously reported that SqrR can bind hemes, and the redox state of the heme iron affects the secondary structure of SqrR [52], suggesting that the SqrR repressor activity can be altered based on the redox state of the heme. However, we note that only 1.7% of SqrR can bind to a heme in vivo under normal growth conditions without abiotic stress. Oxidative stress induces free hemes by dissociation from hemoproteins [53,54], and heme-bound holo-SqrR showed lower DNA-binding affinity than apo-SqrR [52]. Therefore, another possibility is that SqrR senses increased free hemes under oxidative stress. Further analyses are needed to fully understand the mechanism of how SqrR mediates H_2_O_2_ signaling in RcGTA production.

## 5. Conclusions

We suggest that SqrR mediates the H_2_O_2_-induced regulation of RcGTA production by modulating c-di-GMP. Although SqrR appears to regulate RcGTA transcription, the effect is indirect, implicating another regulatory factor of RcGTA that remains unknown. However, our discovery of SqrR as a novel mediator of H_2_O_2_-induced RcGTA production allows for further elucidation of how the whole regulatory network functions in this model GTA-producing bacterium.

## Figures and Tables

**Figure 1 microorganisms-10-00908-f001:**
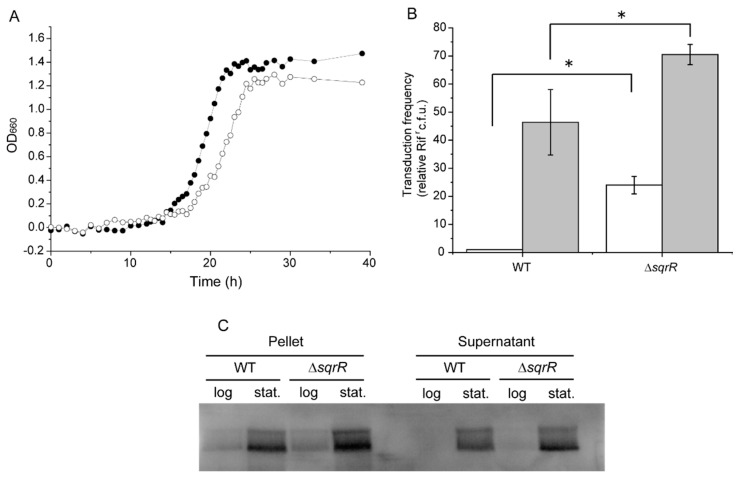
Effects of *sqrR* deletion on growth, gene transfer and GTA production and release. (**A**) Growth curves of the parental wild-type strain (filled circle) and the Δ*sqrR* strain (open circle) of *R. capsulatus* DE442 under anaerobic photosynthetic conditions. Data shown are the mean of triplicate culture tubes. (**B**) Transduction frequencies using GTA produced at the log phase (white bar) and stationary phase (gray bar) in the parental wild-type strain (WT) and the Δ*sqrR* strain. Bars show the mean, error bars show standard deviation of three biological replicates, and star (*) indicates a statistically significant difference (*t*-test, *p*-value < 0.05). (**C**) Western blots of the WT and Δ*sqrR* strain culture cell pellets and supernatant fractions at the log and stationary (stat.) phases, probed with *R. capsulatus* GTA capsid antiserum.

**Figure 2 microorganisms-10-00908-f002:**
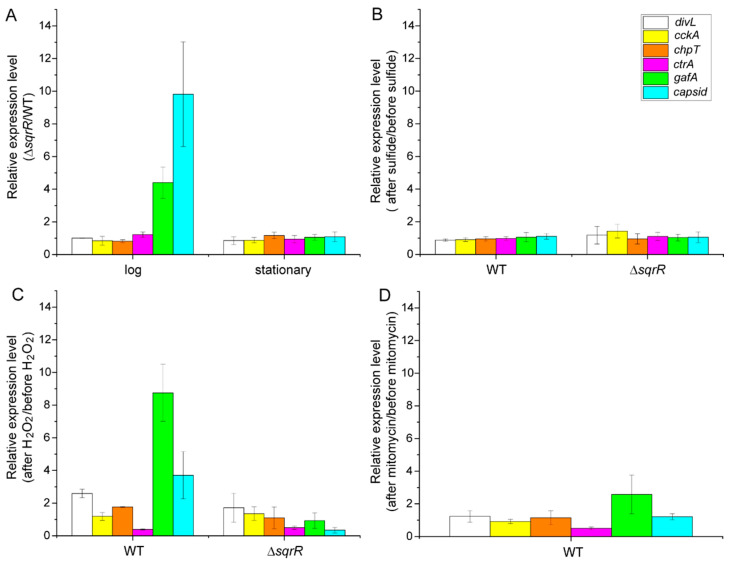
Relative level of transcripts of GTA-related genes. (**A**) Relative expression levels of Δ*sqrR* as compared with the parental wild-type strain (WT) at the log and stationary phases, respectively. Cells were grown under anaerobic photosynthetic conditions. (**B**–**D**) Changes in the relative transcript levels of the GTA-related genes after the addition of 0.2 mM sodium sulfide (**B**), 1 mM hydrogen peroxide (**C**), or 20 µg/mL mitomycin c (**D**) to WT and Δ*sqrR* strains. Cells were grown to mid-log phase under anaerobic photosynthetic conditions and each chemical was added at *t* = 0. Cells were harvested after 30 min and assayed for qRT-PCR. Bars show the mean, and error bars show the standard deviation of the three biological replicates.

**Figure 3 microorganisms-10-00908-f003:**
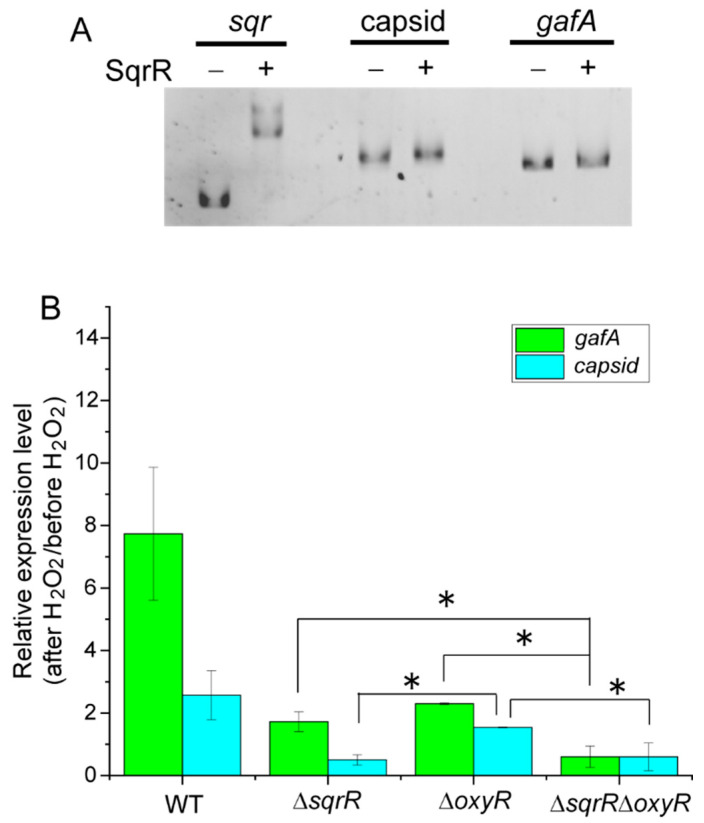
Molecular characterization of SqrR in the H_2_O_2_-induced transcriptional regulation of GTA-related genes. (**A**) Gel mobility shift assay using a DNA probe of the *sqr*, capsid gene and *gafA* promoter region under 5 mM DTT-reducing conditions without (−) or with (+) 0.5 mM DTT-reduced SqrR. (**B**) Changes in the relative transcript levels of the capsid gene (green) and *gafA* (blue) after the addition of 1 mM hydrogen peroxide to WT and each mutant culture. Cells were grown to mid-log phase under anaerobic photosynthetic conditions and each chemical was added at *t* = 0. Cells were harvested after 30 min and assayed for qRT-PCR. Bars show the mean, error bars show standard deviation of three biological replicates, and star (*) indicates a statistically significant difference (*t*-test, *p*-value < 0.05).

**Figure 4 microorganisms-10-00908-f004:**
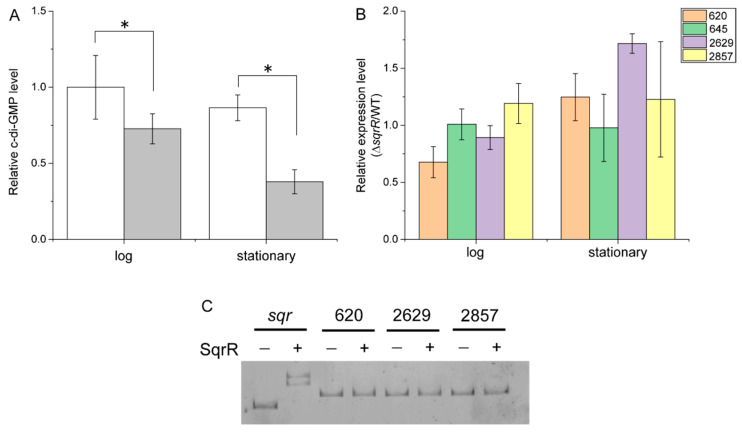
Effects of *sqrR* deletion on the intracellular c-di-GMP levels. (**A**) Relative c-di-GMP levels in the parental wild-type strain (WT) (white bar) and Δ*sqrR* (gray bar) at the log and stationary phases, compared to WT at the log phase. Error bars show the standard deviation of three biological replicates, and star (*) indicates a statistically significant difference (*t*-test, *p*-value < 0.05). (**B**) Relative expression levels in Δ*sqrR,* as compared with WT at the log and stationary phases. Cells were grown under anaerobic photosynthetic conditions. Error bars show standard deviation of three biological replicates. (**C**) Gel mobility shift assay using a DNA probe of the *sqr*, rcc00620, rcc02629 and rcc02857 promoter region under 5 mM DTT-reducing conditions without (−) or with (+) 5 mM DTT-reduced SqrR.

**Figure 5 microorganisms-10-00908-f005:**
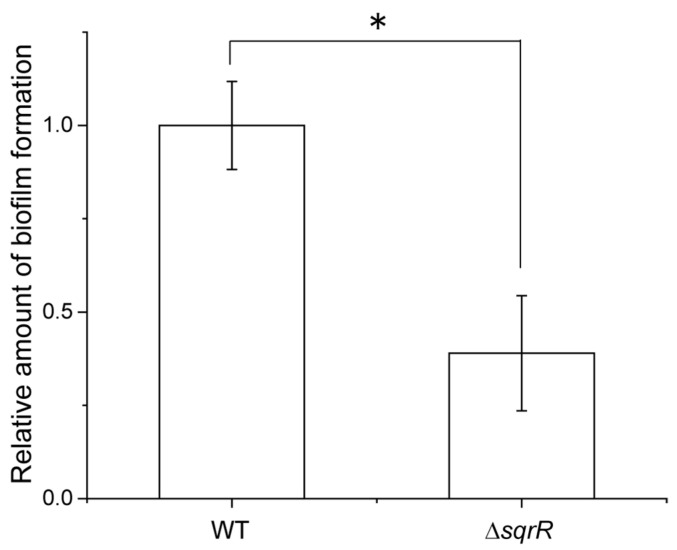
Effects of *sqrR* deletion on biofilm formation. Relative amount of biofilm formation in *sqrR* compared with the parental wild-type strain (WT). Bars show the mean, error bars show the standard deviation of three biological replicates, and star (*) indicates a statistically significant difference (*t*-test, *p*-value < 0.05).

**Table 1 microorganisms-10-00908-t001:** Transcript levels of GTA-related genes affected by the loss of SqrR.

Gene	Annotation	Transcript Fold Change in Δ*sqrR*	*p*-Value
rcc00042	divL	9.276648	4.94 × 10^−64^
rcc01687	capsid	3.318318	2.28 × 10^−7^
rcc03000	chpT	2.984249	2.6 × 10^−8^
rcc00645	c-di-GMP metabolic enzyme	118.6299	3.7 × 10^−167^
rcc02857	c-di-GMP metabolic enzyme	0.497602	0.002762

**Table 2 microorganisms-10-00908-t002:** The list of primers used in this study.

Name	Sequence 5′–3′	Purpose
oxyR F1	CGACTCTAGAGGATCATTGCCGTATTTCTTCTTGATCGGC	Cloning for gene disruption
oxyR R1	CGAGAGGTTTATCATAATGAAAAACTATCGCAGGC	
oxyR F2	ATGATAAACCTCTCGGCGCGGGAGGCGTGAGGTCGGCGGGTTCGG	
oxyR R2	CGGTACCCGGGGATCGCGCATCCGCTGGCGCCCGAGACGC	
divL-F	CGGTACCCGGGGATCAGAATGCGCCGGTGCCGCGGCGCTC	Cloning for DNA probes of the gel shift assay
divL-R	CGACTCTAGAGGATCGACTGCAGCCCTCTCGCCTGTCCCG	
cckA-F	CGGTACCCGGGGATCCGCCGCAGCTATTCCCCGCGCGACG	
cckA-R	CGACTCTAGAGGATCGGGCTGATCGGGATGTACCACTGGC	
chpT-F	CGGTACCCGGGGATCAAGCTGCACCCGTCGCCCGTCGATC	
chpT-R	CGACTCTAGAGGATCGGGTCATGGTGGATCTCCCTTTCGG	
gafA-F	CGGTACCCGGGGATCGTAATCGCGCTGCCCGAAGCGTGCG	
gafA-R	CGACTCTAGAGGATCCTCCGGTCTCCCATCGACAGGCTGG	
capsid-F	CGGTACCCGGGGATCACCGGCGGGCATGCTTTTGCCGAGA	
capsid-R	CGACTCTAGAGGATCGTCTTGCGTGACCCGCCTCTCATGC	
ctrA-F	CGGTACCCGGGGATCGCCGCCGAAAGAAACGCGTCGTTGG	
ctrA-R	CGACTCTAGAGGATCCCTGGGTTCTCCGCATTAATCCCTC	
02857-F	CGGTACCCGGGGATCTGGTGCCCCAGCCTAACCGCGGGAT	
02857-R	CGACTCTAGAGGATCCGGGAACGGACCCCTTCGAGTGGAT	
02630-F	CGGTACCCGGGGATCGTGCCCGGACCGGAGGCGGTTTTCC	
02630-R	CGACTCTAGAGGATCGGACCCTCCTCGCGCGGACCATAGC	
00620-F	CGGTACCCGGGGATCCTTGTCGGGGGGGATGACGCCGCTG	
*uvrD* qF	CAGAAGGAACACACGGTCAA	For qRT-PCR
*uvrD* qR	AAAGTGTCAGGCGGAATCTC	
*divL* qF	CCGACGCTTTATGCCTTTCT	
*divL* qR	CCTGTTCCAGTTCCGTCATCT	
*cckA* qF	GCGCATGATTTCAACAACTT	
*cckA* qR	TTCTGGCTGATCTGGTCAAG	
*chpT* qF	ACGGGGTGGAGTTGCTGAA	
*chpT* qR	AAAGGCGATGCGGAAGAA	
*ctrA* qF	TTTGGCGCCGATGATTAC	
*ctrA* qR	GGATGATCGACTGCGAATG	
*gafA* qF	GCTGAACGGCTGGATCTT	
*gafA* qR	TTCCAACAGCCGCTTCAA	
capsid qF	CGGTTGCCGAGGTGAAA	
capsid qR	CACACGCTCTTCCTGTTGTTG	

## Data Availability

Not applicable.

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
