# Peer review of "Persulfide-Responsive Transcription Factor SqrR Regulates Gene Transfer and Biofilm Formation via the Metabolic Modulation of Cyclic di-GMP in Rhodobacter capsulatus"

_microorganisms, 2022, doi:10.3390/microorganisms10050908_

Round 1

Reviewer 1 Report

The manuscript describes a persulfide-responsive transcription factor SqrR that is also responsive to H2O2, mediating the regulation of GTAs and biofilm development via c-di-GMP. The in vivo evidence is supportive, but the in vitro results did not reach any new conclusions.  A sensing mechanism is proposed. H2O2 treatment may lead to elevated levels of intracellular heme, and SqrR binds a heme to regulate c-di-GMP production.  Further studies are necessary to verify the hypothesis.

Specific comments

Line 2, 13, 70, 230: Be consistent: persulfide-responsive (instead of persulfide responsive or (per)sulfide-responsive) SqrR throughout the manuscript.

Lines 119 and 120: Check the volume uL  or mL?

Line 380: the heme-binding ability of SqrR

Line 381: SqrR can bind a heme.

Author Response

We thank both reviewers for their constructive comments. In the revised manuscript, appended or modified sentences are marked up using the “Track Changes” function according to the instruction from the MDPI editorial office.

Reply to Reviewer 1

Comments to the Author

The manuscript describes a persulfide-responsive transcription factor SqrR that is also responsive to H2O2, mediating the regulation of GTAs and biofilm development via c-di-GMP. The in vivo evidence is supportive, but the in vitro results did not reach any new conclusions.  A sensing mechanism is proposed. H2O2 treatment may lead to elevated levels of intracellular heme, and SqrR binds a heme to regulate c-di-GMP production.  Further studies are necessary to verify the hypothesis.

OUR ANSWER: We thank the reviewer for the valuable comments and recommendations. We agree with reviewer’s comment “Further studies are necessary to verify the hypothesis.” Because we had written, “Further analyses are needed to fully understand the mechanism of how SqrR mediates H2O2 signaling in RcGTA production” in the original manuscript (Lines 447-449 in the revision), we did not make any change in response to this comment. Scientific advances typically progress incrementally, and it is rare that a single publication has all the answers to a particular problem.

Specific comments

Line 2, 13, 70, 230: Be consistent: persulfide-responsive (instead of persulfide responsive or (per)sulfide-responsive) SqrR throughout the manuscript.

OUR ANSWER: We thank the reviewer for pointing out lack of unity of the words. We unified to “persulfide-responsive” throughout the manuscript.

Lines 119 and 120: Check the volume uL or mL?

OUR ANSWER: We thank the reviewer for noticing the mistake. We changed the font to “Symbol”.

Line 380: the heme-binding ability of SqrR

Line 381: SqrR can bind a heme.

OUR ANSWER: We revised the points as suggested.

Reviewer 2 Report

The manuscript reports that the GTA activity is negatively regulated by SqrR. The induction of H2O2 to GTA activity is related to SqrR. In the following steps, the authors found that SqrR could regulate GTA activity by affecting intracellular cyclic di-GMP levels. The biofilm formation was also reduced due to the decrease of cyclic di-GMP in ΔsqrR.

In the previous studies on this kind of regulator proteins, the results are related with the regulation on sulfur metabolism. This paper tried to think out of the box and explored new targets. The biofilm formation ability and the HGT activity were found to be related with this regulator. The authors also tried to illustrate the mechanism.  Their findings are interesting that will be suitable for publication in Microorganisms if following comments could be addressed:

  1. Line 268-272: It is not clear if these results supported the independency of SqrR and OxyR. I suggested the authors could show us the fold changes of indicated groups and calculated the significant differences.
  2. In Fig. 2C and Fig. 3B, the transcription levels of gafA in Δsqr cells are different from what.
  3. Line 28:The full name of HGT should be given.
  4. Line 92, Line 107, Line 119 and so on, the symbol in unit is missing.
  5. Line 142: 0.5 mL
  6. Line 198: log, not lag

Author Response

We thank both reviewers for their constructive comments. In the revised manuscript, appended or modified sentences are marked up using the “Track Changes” function according to the instruction from the MDPI editorial office.

Reply to Reviewer 2

Comments to the Author

The manuscript reports that the GTA activity is negatively regulated by SqrR. The induction of H2O2 to GTA activity is related to SqrR. In the following steps, the authors found that SqrR could regulate GTA activity by affecting intracellular cyclic di-GMP levels. The biofilm formation was also reduced due to the decrease of cyclic di-GMP in ΔsqrR.

In the previous studies on this kind of regulator proteins, the results are related with the regulation on sulfur metabolism. This paper tried to think out of the box and explored new targets. The biofilm formation ability and the HGT activity were found to be related with this regulator. The authors also tried to illustrate the mechanism.  Their findings are interesting that will be suitable for publication in Microorganisms if following comments could be addressed:

1. Line 268-272: It is not clear if these results supported the independency of SqrR and OxyR. I suggested the authors could show us the fold changes of indicated groups and calculated the significant differences.

OUR ANSWER: We thank the reviewer for the valuable comments and recommendations. We added the information about significant differences in Fig. 3B and describe the significant differences in the text (Lines 325-328).

2.In Fig. 2C and Fig. 3B, the transcription levels of gafA in Δsqr cells are different from what.

OUR ANSWER: We performed each experiments in Fig. 2C and 3B; therefore, the transcript levels of gafA in DsqrR cells were a little changed. However, there were no significant differences between the data in Fig. 2C and 3B. Moreover, in both Fig. 2C and 3B, it is clear that the transcript levels of gafA in ΔsqrR cells were different from the levels in WT cells. Therefore, we did not make any change in response to this comment.

3. Line 28:The full name of HGT should be given.

OUR ANSWER: We thank the reviewer’s pointing out. We now give the full name of HGT.

4. Line 92, Line 107, Line 119 and so on, the symbol in unit is missing.

5. Line 142: 0.5 mL

6. Line 198: log, not lag

OUR ANSWER: We thank the reviewer for noticing the mistakes. We revised the points as suggested.